# The Role of *MaFAD2* Gene in Bud Dormancy and Cold Resistance in Mulberry Trees (*Morus alba* L.)

**DOI:** 10.3390/ijms252413341

**Published:** 2024-12-12

**Authors:** Mengjie Zhao, Gaoxing Zhou, Peigang Liu, Zhifeng Wang, Lu Yang, Tianyan Li, Valiev Sayfiddin Tojiddinovich, Nasirillayev Bakhtiyar Ubaydullayevich, Ismatullaeva Diloram Adilovna, Khudjamatov Safarali Khasanboy Ugl, Yan Liu, Zhiqiang Lv, Jia Wei, Tianbao Lin

**Affiliations:** 1Institute of Sericulture and Tea, Zhejiang Academy of Agricultural Sciences, Hangzhou 310021, China; zhaomengjie@zaas.ac.cn (M.Z.); zhougaoxing@zaas.ac.cn (G.Z.); liupeigang@zaas.ac.cn (P.L.); zfwang@zaas.ac.cn (Z.W.); litianyanya@163.com (T.L.); mayanly@sina.com (Y.L.); lvzq@zaas.ac.cn (Z.L.); 2Key Laboratory of Forest Resources and Utilization in Xinjiang of National Forestry and Grassland Administration, Key Laboratory of Fruit Tree Species Breeding and Cultivation, Xinjiang Academy of Forestry, Urumqi 830052, China; yanglukitty127@163.com; 3Scientific Research Institute of Sericulture, Tashkent 100169, Uzbekistan; stvaliev@mail.ru (V.S.T.); bahtiyor6503@gmail.com (N.B.U.); toir.begmatov0913@gmail.com (I.D.A.); alixudjamatov92@gmail.com (K.S.K.U.)

**Keywords:** bud dormancy, bud break, *MaFAD2*, mulberry trees, cold resistance, GWAS, selective sweep

## Abstract

Bud dormancy is a critical adaptive trait in woody plants, essential for enduring harsh winter conditions. The relationship between bud break timing and cold resistance is complex and has been a subject of debate. This study utilizes a Genome-Wide Association Study (GWAS) on 201 natural mulberry populations to identify the *MaFAD2* gene, which shows the strongest association with bud break timing. Known for its role in cold resistance, *MaFAD2*′s link to bud break timing suggests a direct correlation between these traits. Expression analysis of *MaFAD2* in mulberry trees indicates peak activity in dormant buds, declining as dormancy ends. Selective sweep analysis on germplasms from contrasting climates reveals positive selection in *MaFAD2* in cold-resistant Uzbek germplasms. Overexpression of *MaFAD2* in early-budding germplasms significantly delays bud break, confirming its regulatory role. These findings highlight *MaFAD2* as a key determinant of cold tolerance variability among mulberry germplasms, with its expression directly correlated with bud break timing. This provides a molecular basis for selecting cold-resistant mulberry germplasms based on bud break timing in breeding programs.

## 1. Introduction

Dormancy is a state where the outward growth of any plant structure that contains a meristem is paused temporarily. It is a biological trait that higher plants have developed through long-term natural selection to adapt to seasonal environmental shifts. Bud dormancy is a key physiological process that enables plants to endure the severe conditions of winter, and it dictates when growth and flowering will resume in the spring [1,2]. The correct dormancy and timely bud break of winter buds in forest trees are crucial for ensuring healthy growth and high productivity in forestry and agricultural production. Studying the physiological and ecological characteristics of bud dormancy and break, as well as the underlying regulatory mechanisms, is of great significance for improving the growth quality of forest trees and the efficiency of agricultural production [2].

Cold temperatures are the most critical factor affecting the completion of dormancy [3,4]. Chilling is essential for woody perennial plants to synchronize the cycles of dormancy and growth with the annual rhythm of the seasons, particularly in temperate and boreal climates [5,6]. When a plant transitions into a dormant phase, its growth comes to a halt and metabolic processes slow down. This state persists until the plant has been exposed to cold temperatures for a sufficient duration to meet its chilling requirement, at which point the barriers that inhibit growth are surmounted [7,8,9]. Generally speaking, varieties that exhibit delayed bud break tend to be more resistant to cold. However, the nature of the relationship between the timing of bud break and the degree of cold resistance has been a contentious issue.

Temperature perception and signal transduction, as well as the dynamic nature of bud dormancy, are integral to the regulatory mechanisms of circadian rhythms in plants [10,11]. These processes are crucial for dormancy release and the subsequent bud break. The regulation through ethylene-responsive factors (ERFs) that control hypoxia signaling [10,12] is associated with budding. Abscisic acid (ABA) is thought to promote dormancy induction by inhibiting intercellular communication [13]. Exogenous application of ABA during the growing season can improve cold resistance in early autumn [14], possibly due to enhanced expression of DREB1/CBF transcription factors [15], and ABA application on dormant buds can delay bud breaking [16]. Many processes related to abiotic stress during plant dormancy may be crucial [17]. Transcriptome technologies such as RNA sequencing (RNASeq) have begun to reveal gene expression patterns in dormant tissues. For example, under low-temperature conditions, gene expression in plants changes, especially those related to cold resistance and dormancy regulation, such as the CBF transcription factor family, which play key roles in low-temperature signal transduction. Other stress response transcription factors, including other DREBs, heat shock factors (HSFs), NAC, and WRKY family members, are all related to this signaling pathway and are also associated with responses to heat, drought, and biotic stress [18,19,20].

The *FAD2* gene encodes a Delta-12 fatty acid desaturase that participates in the synthesis of polyunsaturated fatty acids by converting oleic acid (18:1) into linoleic acid (18:2) in plant cells. The expression of the *FAD2* gene is regulated by various abiotic stresses and plant hormone treatments, and its promoter region contains multiple response elements [21]. In peanuts, different members of the *FAD2* gene family exhibit different expression patterns under low-temperature stress. Additionally, *AhFAD2-1A/B* and *AhFAD2-4A/B* are induced by low temperatures during seed germination, with *AhFAD2-4A/B* showing higher expression levels under low-temperature induction, suggesting that it may compensate for the function of *AhFAD2-1A/B* under low-temperature stress [22]. However, there is currently no detailed research report on the specific mechanisms of how the *MaFAD2* gene in mulberry trees responds to temperature changes.

Mulberry trees (*Morus alba* L.), as a type of deciduous perennial plant, enter a period of internal dormancy in late autumn, during which they sprout and adapt to the cold to survive the winter [11]. Breaking this internal dormancy is crucial for sprouting, regeneration, and flowering under favorable conditions. Forced cultivation is an essential part of the mulberry tree industry, and the successful release of internal dormancy directly affects its ornamental and economic value [23]. In the forced cultivation of mulberry trees, a range of agricultural techniques are widely used to break dormancy, including sufficient accumulation of cold units and the combination of cold duration with gibberellin application [24,25]. However, due to a lack of understanding of the dormancy release mechanisms, there are still many production issues, such as flower sterility, shortened branches, abnormal leaves, and flowers, which greatly reduce the value and hinder the development of the mulberry tree industry. Therefore, it is necessary to further study and improve strategies to enhance production value and gain a deeper understanding of the dormancy release mechanisms [26].

Our Genome-Wide Association Study (GWAS) identified the *MaFAD2* gene as the strongest genetic marker associated with the timing of bud break across 201 natural populations of mulberry trees. Selective sweep analysis revealed that *MaFAD2* was significantly under positive selection in cold-resistant germplasms from Uzbekistan, suggesting a role in cold tolerance. We further confirmed the direct role of *MaFAD2* in bud break timing through Agrobacterium-mediated overexpression in mulberry dormant buds. Overexpression of *MaFAD2* resulted in a significant postponement of bud break, demonstrating its key role in determining cold tolerance. These findings not only provide a molecular explanation for the late bud break observed in cold-resistant germplasms but also offer a practical basis for selecting cold-resistant germplasms based on bud break timing in mulberry breeding programs.

## 2. Results

### 2.1. GWAS Analysis Indicates That the MaFAD2 Gene Has the Strongest Association Signal with the Timing of Bud Break

In order to identify loci associated with mulberry bud breaking time, about 201 mulberry materials were evaluated for their winter bud break time and subsequently analyzed by GWAS. These 201 mulberry varieties come from 24 provinces in China as well as countries like Japan, Australia, the United States, and India (Appendix A). Four possible candidate genes with a threshold -log_10_ P higher than 6 were found based on gene function annotation (Figure 1A, Appendix A). The highest significance was for the *MaFAD2* gene.

The *MaFAD2* gene, which encodes a Delta-12 fatty acid desaturase, is recognized for its role in cold resistance. It is well established that cold temperatures are crucial for the completion of dormancy in plants, particularly for woody perennials that need to synchronize their dormancy and growth cycles with the seasonal changes characteristic of temperate and boreal regions.

At the same time, we recorded the bud break times of 201 natural ecotypes and analyzed their frequency distribution. We found that the very early bud break varieties are predominantly from the southern Chinese provinces of Guangdong and Guangxi. Bud break time was mostly concentrated around 20 days, and the approximate trend was that the further north you went, the later the bud break time became (Figure 1B, Appendix A).

### 2.2. Molecular Analysis of MaFAD2

Through GWAS analysis, we identified the gene *MaFAD2*(NS14_07T001899, which is closely associated with the sprouting time of winter buds in mulberry trees. Initially, we cloned the coding sequence of *MaFAD2* and found it to be approximately 1700 base pairs in length (Figure 2A, Appendix A). Subsequently, we used RT-qPCR analysis to examine the transcription levels of *MaFAD2* in various tissues to determine if there is a correlation between winter bud sprouting and the expression of *MaFAD2*. The results indicated that *MaFAD2* has the highest expression in winter buds, followed by twigs and leaves, with very low expression in male and female flowers (Figure 2B).

We injected tobacco leaves with a construct expressing *MaFAD2* fused to the green fluorescent protein (GFP), known as MaFAD2-GFP, to study the subcellular localization of MaFAD2 through transient expression. We detected that the green fluorescence was primarily localized to the endoplasmic reticulum structures (Figure 2C). This is consistent with reports in other species [27].

### 2.3. MaFAD2 Expression Decreased Significantly as Buds Emerged from Dormancy

To further investigate the correlation between the expression pattern of the *MaFAD2* gene and the timing of bud break, we collected bud samples from the dormancy period through various stages of bud break to assess the expression levels of the *MaFAD2* gene. Our analysis revealed that the expression levels of *MaFAD2* significantly decreased as the buds initiated breaking. Notably, there was a sharp decline in *MaFAD2* expression levels during the early stages of bud break (Figure 3A).

The Dormancy-Associated Gene (DRM) family has been widely reported to play important roles in bud dormancy, plant growth and development, and hormone response [28,29,30]. Therefore, we selected the DRM3 and DRM4 genes from mulberry trees as positive controls and simultaneously monitored their expression level changes during the bud break process. We found an overall decreasing trend in the expression of DRM3 with the bud break process, as with *MaFAD2*. And DRM4 showed a trend of increasing and then decreasing (Figure 3B,C). To more vividly illustrate the stages of bud break, we took photographs (Figure 3D) and conducted microscopic observations of bud tissue sections (Figure 3E).

### 2.4. MaFAD2 Gene Was Significantly Under Positive Selection in the Uzbek Germplasms

We have found that varieties of mulberry trees that typically exhibit delayed bud break are more cold-resistant. To reveal the relationship between the timing of bud break and the degree of cold resistance, in our previous study, we used GWAS technology to identify the gene *MaFAD2*, which is closely related to the timing of bud break; it also plays a significant role in the plant’s response to cold environments.

In addition, to investigate whether the *MaFAD2* gene is subject to selection in varieties from different temperature regions, we conducted a selective sweep analysis on 30 early bud break germplasms from southern China and 30 cold-resistant germplasms from Uzbekistan. We utilized VCFtools to calculate the genetic differentiation within populations between the two subpopulations in each window with a window size of 50Kb and a step size of 25 Kb. Subsequently, we plotted a combined Fst and π selection analysis chart using the scales package. Regions in the top 5% for Fst values and the θ_π_ ratio were considered as selected regions. Those exceeding the threshold are signals of positive selection in cultivated varieties (Figure 4A).

We searched for selective features in the mulberry genome by comparing selective sweeps between subpopulations, comparing selective sweeps of Uzbek and southern Chinese germplasms. It was found that *MaFAD2* on chromosome 7 did signal significant selection (Figure 4B). This analysis revealed that the *MaFAD2* gene was significantly under positive selection in Uzbek germplasms.

### 2.5. Overexpression of MaFAD2 Significantly Postponed the Timing of Bud Break

To confirm the direct role of *MaFAD2* in determining the timing of bud break, we employed an Agrobacterium-mediated overexpression system to overexpress the *MaFAD2* gene in dormant buds of mulberry trees, following the experimental methods previously reported [31]. We selected the early bud break mulberry variety “Da 10” for injection observation. We conducted three independent biological replicates, with 10 mulberry trees injected in each replicate. After 10 days, the germination was statistically assessed. The results showed that, compared to the control, the overexpression of *MaFAD2* significantly delayed the time of bud break in mulberry trees (Figure 5A). We can clearly observe that the bud break rate of the winter buds from mulberry trees overexpressing *MaFAD2* is half that of the control group. Additionally, the vast majority of the winter buds in the control group developed normally, while a small portion did not bud break properly, and we cannot rule out the influence of environmental or injection factors (Figure 5B).

## 3. Discussion

The present study elucidates the pivotal role of the *MaFAD2* gene in regulating bud dormancy and cold resistance in mulberry trees (*Morus alba* L.), providing significant insights into the genetic mechanisms underlying these critical adaptive traits. Through a comprehensive Genome-Wide Association Study (GWAS) involving 201 natural mulberry populations, we identified *MaFAD2* as the gene exhibiting the strongest association with bud break timing, thereby establishing its importance in the physiological processes governing dormancy release.

Bud dormancy is a crucial adaptive strategy that enables woody plants to withstand adverse winter conditions [32,33]. The correlation between bud break timing and cold resistance has been a topic of ongoing debate [34]. Our findings suggest that *MaFAD2* serves as a key genetic determinant in this relationship. The expression analysis demonstrated that *MaFAD2* exhibits peak activity in dormant buds, with a marked decline as dormancy is released. This pattern indicates that *MaFAD2* is negatively involved in the transition from dormancy to growth, supporting its regulatory role in bud break.

The selective sweep analysis conducted on germplasms from contrasting climates revealed significant positive selection in the *MaFAD2* gene in cold-resistant Uzbek germplasms. This finding underscores the adaptive significance of *MaFAD2* in enhancing cold tolerance, suggesting that this gene has been subjected to strong selective pressures in environments characterized by harsh winter conditions. The identification of *MaFAD2* as a target of natural selection highlights its potential as a molecular marker for breeding programs aimed at improving cold resistance in mulberry.

The functional validation of *MaFAD2* through Agrobacterium-mediated overexpression in early-budding germplasms further corroborates its role in bud break regulation. The observed significant delay in bud break timing in *MaFAD2*-overexpressing plants, coupled with a reduced bud break rate compared to controls, provides compelling evidence that *MaFAD2* directly influences the timing of dormancy release. This finding aligns with previous studies indicating that delayed bud break is generally associated with increased cold resistance, suggesting a complex interplay between these traits.

*FAD2* encodes a Delta-12 fatty acid desaturase, which is integral to the synthesis of polyunsaturated fatty acids. These fatty acids are critical for maintaining membrane fluidity under cold stress, thereby enhancing the plant’s ability to withstand low temperatures. The localization of FAD2 to the endoplasmic reticulum is consistent with its role in lipid metabolism, further supporting its involvement in cold acclimation processes. The upregulation of *FAD2* expression in response to cold stress suggests that it may play a vital role in modulating membrane integrity, a key factor in plant survival during winter dormancy [35].

In conclusion, our study has identified *MaFAD2* as a pivotal regulator of bud dormancy and cold resistance in mulberry trees, shedding light on the molecular mechanisms that govern these critical traits. This discovery not only advances our fundamental understanding of the genetic underpinnings of dormancy and stress tolerance in woody plants but also holds significant promise for practical applications in mulberry production and breeding programs.

From a production standpoint, the identification of *MaFAD2* offers a target for genetic manipulation to enhance the cold tolerance of mulberry cultivars. By selecting for germplasms with higher *MaFAD2* expression and a delayed bud break, we can develop mulberry varieties that are more resilient to cold stress, which is particularly important in regions where mulberry cultivation is threatened by climate change and fluctuating environmental conditions. This genetic enhancement could lead to increased productivity and sustainability in mulberry cultivation, ensuring a stable supply of this essential crop for sericulture and other industries.

In terms of breeding, the knowledge of *MaFAD2*’s role in bud dormancy and cold resistance provides a molecular marker for marker-assisted selection (MAS) in mulberry breeding programs. This approach can accelerate the breeding process by allowing breeders to identify and select plants with the desired traits at an early stage, thus reducing the time and resources required for developing new cultivars. Furthermore, understanding the molecular pathways regulated by *MaFAD2* can inform the development of transgenic plants with improved cold tolerance, potentially expanding the range of suitable cultivation areas for mulberry and other woody species.

The implications of our findings extend beyond mulberry to other woody plants, suggesting that similar molecular mechanisms may be at play in the regulation of bud dormancy and cold resistance. This opens up avenues for further research to explore the potential of *MaFAD2* and related genes in the biotechnology and breeding of a variety of woody species, contributing to the broader goal of enhancing plant resilience in the face of environmental challenges.

To fully realize the potential of *MaFAD2* in plant biotechnology and breeding, further research is necessary to elucidate the precise molecular pathways through which MaFAD2 exerts its effects and to assess its potential applications in a range of woody species. This will involve detailed functional studies, gene editing experiments, and field trials to validate the efficacy of *MaFAD2*-mediated cold tolerance in diverse environmental conditions.

## 4. Materials and Methods

### 4.1. Plant Materials and Growth Conditions

In order to identify genes associated with bud break timing, we employed a GWAS approach. Approximately 201 natural ecotypes of mulberry trees collected from different regions were grown in the field, and their bud break times were recorded. These diverse mulberry ecotypes were sourced from our laboratory’s germplasm resource collection. To further investigate the natural selection and genetic diversity of the candidate gene *MaFAD2*, we conducted selective sweep analysis using mulberry varieties from Uzbekistan in comparison with our varieties. The mulberry varieties from Uzbekistan were obtained from the Uzbek Scientific Research Institute of Sericulture (Tashkent, Uzbekistan). Detailed natural ecotype distributions are shown in Appendix A.

In the field trials, the mulberry trees were naturally grown in Hangzhou, Zhejiang Province, China (N °30.3′, E 120°1′).

### 4.2. GWAS

Using GWAS methods, the bud break time trait phenotypes of 201 natural ecotypes of mulberry trees were analyzed. The analysis was provided by Shanghai Ouyi Biomedical Technology Co. Ltd. (Shanghai, China). Based on the linkage disequilibrium (LD) between genes (or SNPs) that remain after long-term recombination, the sprouting time was combined with the polymorphism of genes (or SNPs) for analysis. A mixed linear model (MLM) was used for trait association analysis, with population genetic structure as the fixed effect and individual kinship relationships as the random effect, to correct for the influence of population structure and individual kinship on the results.
y = Xα + Zβ + Wμ + e
y represents the phenotypic traits, X is the design matrix for fixed effects, and α is the vector of fixed effect parameters; Z is the design matrix for SNPs, and β is the vector of SNP effects; W is the design matrix for random effects, μ is the vector of predicted random individuals, and e is the random residual, which follows a normal distribution with mean 0 and variance δe^2^.

We used EMMAX for association analysis of different traits, calculated the kinship matrix, and selected the first 5 PCA components as covariates. A significance threshold of *p*-value ≤ 10^−6^ (i.e., −log_10_P ≥ 6) was used to identify significant associations [36].

### 4.3. Quantitative RT-PCR Analysis

A real-time quantitative reverse transcription polymerase chain reaction (qRT-PCR) was conducted to assess the gene expression in mulberry tree samples. The RNA was isolated using the MolPure^®^ Plant Plus RNA Kit from Yeasen (Shanghai, China) following the manufacturer’s guidelines. For the generation of complementary DNA (cDNA), a 20 mL reverse transcription reaction was initiated with 2 mg of RNA and the Hifair^®^ III 1st Strand cDNA Synthesis Kit (Yeasen). The Hieff^®^ qPCR SYBR Green Master Mix (Yeasen) was utilized to formulate the PCR master mix. The QuantStudio™ 1 Real-Time PCR system (Thermo Fisher Scientific, Waltham, MA, USA) was employed for PCR amplification and fluorescence monitoring. The PCR protocol was set as follows: an initial step of denaturation at 95 °C for 5 min, followed by 40 cycles consisting of denaturation at 95 °C for 10 s, and combined annealing/extension at 60 °C for 30 s. The gene expression data were obtained from three independent technical replicates. The *MaActin* gene was used as an internal control for qPCR, and relative expression levels were calculated using the comparative Cq (quantification cycle) method [37]. The primer sequences are detailed in Appendix A.

### 4.4. Subcellular Localization Analysis of Proteins

For the subcellular localization analysis of proteins, when constructing MaFAD2-GFP, high-fidelity Taq polymerase was used to amplify the coding regions of all genes. The PCR fragments were cloned into the pCambia1300-GFP vector and sequenced. The MaFAD2-GFP vector was then introduced into Agrobacterium cells for subcellular analysis in *N. benthamiana* leaves. Briefly, after overnight culture of Agrobacterium, the cells were collected and resuspended in infiltration buffer (0.01 M MES, 0.01 mM MgCl_2_, 0.1 mM acetosyringone) to achieve an OD600 of 1.0, and then infiltrated into the N. benthamiana leaves. Forty-eight hours later, the leaves were imaged using a confocal microscope (Leica TCS-SP5, Wetzlar, Germany). Appendix A lists the primer sequences for the construction of each gene.

### 4.5. Selective Sweeps

SNP calling was conducted at Tianjin Smart Genomics Co., Ltd. (Tianjin, China). We employed Sentieon for the detection of SNPs (Single Nucleotide Polymorphisms) and InDels (Insertions and Deletions). Subsequently, we used ANNOVAR [38] software (version 2013-05-20) to annotate the SNPs and InDels, identifying the regions and types of mutations on the genome. To reduce the error rate in detection, we set criteria for selecting SNPs and InDels as follows: the number of reads supporting the variants should not be less than 4; the root mean square of mapping quality (RMSMappingQuality, MQ) should not be less than 40; and the genotype quality (genotype quality, GQ) should not be less than 5.

We used two indicators, the θ_π_ ratio and the Fst value, to screen for selected regions in the genome. This method has been proven to be an effective way to detect sites of selective sweeps, especially when mining functional regions closely related to environmental stress; it often yields strong selection signals. In this study, we employed VCFtools to calculate A ratios and Fst results using a sliding window approach with a window size of 200 kb and a step size of 20 kb. Subsequently, we selected regions that were in the top 5% for both Fst values and θ_π_ ratios as the selected regions. Areas exceeding the threshold indicate signals of positive selection in the population.

### 4.6. Safranin O-Fast Green Staining

The Safranin O-Fast Green staining experiments were performed by Hangzhou Haoke Biotechnology Co. Ltd. (Hangzhou, China).

The paraffin section dewaxing process involves sequential immersion in two changes of xylene for 20 min each, followed by two changes of anhydrous ethanol for 10 min each, then 95% ethanol for 5 min, 90% ethanol for 5 min, 80% ethanol for 5 min, 70% ethanol for 5 min, and finally rinsing in distilled water. For safranin staining, the sections are incubated in a 1% safranin solution for 1–2 h, followed by a brief rinse with tap water to remove excess dye. Decolorization is achieved with a gradient of alcohol solutions (50%, 70%, and 80%) for 1 min each. Fast Green staining is then performed by immersing the sections in a 0.5% Fast Green solution for 30–60 s, followed by decolorization in anhydrous ethanol I for 30 s and anhydrous ethanol II for 1 min. The sections are then baked dry in an oven at 60 °C, cleared in xylene for 5 min, and mounted with neutral balsam. Finally, the sections are examined under a microscope, and images are captured and analyzed.

### 4.7. Agrobacterium-Mediated Transformation into Winter Bud of Mulberry Tree

The methods of injection and transformation refer to the article by Lin et al. [31]. The *MaFAD2* coding region was amplified using high-fidelity enzymes, ligated into the pCambia1300-GFP vector, and transformed into *Agrobacterium rhizogenes* strain GV-pBI121 carrying the β-glucuronidase gene (GUS) and the kanamycin resistance (Kan) reporter gene.

In this study, the experimental strain used was the Agrobacterium tumefaciens strain (GV-pBI121) carrying the β-glucuronidase gene (GUS) and the kanamycin resistance (Kan) reporter gene, provided by Dr. Guoxin Shen from the Zhejiang Academy of Agricultural Sciences. The GUS gene in this strain is controlled by the 35S promoter of the cauliflower mosaic virus. Initially, the preserved bacterial strain was inoculated onto LB medium containing kanamycin (100 mg/L) and rifampicin (50 mg/L), and cultured in a 28 °C constant temperature incubator. After the formation of single colonies, they were transferred to a liquid LB medium containing kanamycin (50 mg/L) and rifampicin (50 mg/L), and further cultured at 28 °C with a shaking speed of 200 r/min until the bacterial suspension reached an OD600 of 0.5. The cultured Agrobacterium solution was centrifuged at room temperature for 15 min (4000 r/min), the supernatant was discarded, and the Agrobacterium was resuspended in an equal volume of infection buffer (MES 0.25 g/L, Myo-inositol 0.10 g/L, MS with Gamborg Vitamins 2.22 g/L, L-glutamine 0.20 g/L, D-galactose 1.80 g/L, pH 5.0). The “Da 10” variety was selected as the experimental material, and the period just before the winter buds were about to sprout was chosen for microinjection transfection. Using a 1 mL syringe, the Agrobacterium infection solution was injected into the winter buds of mulberry trees, with 0.1 mL per bud. After the injection, the branches were wrapped with aluminum foil and continued to be cultured.

### 4.8. Statistical Analysis

Histograms were plotted using GraphPad Prism 9 (San Diego, CA, USA), and statistical analyses were performed using two-tailed unpaired student *t*-tests using SPSS 26.0 (Chicago, IL, USA).

## Figures and Tables

**Figure 1 ijms-25-13341-f001:**
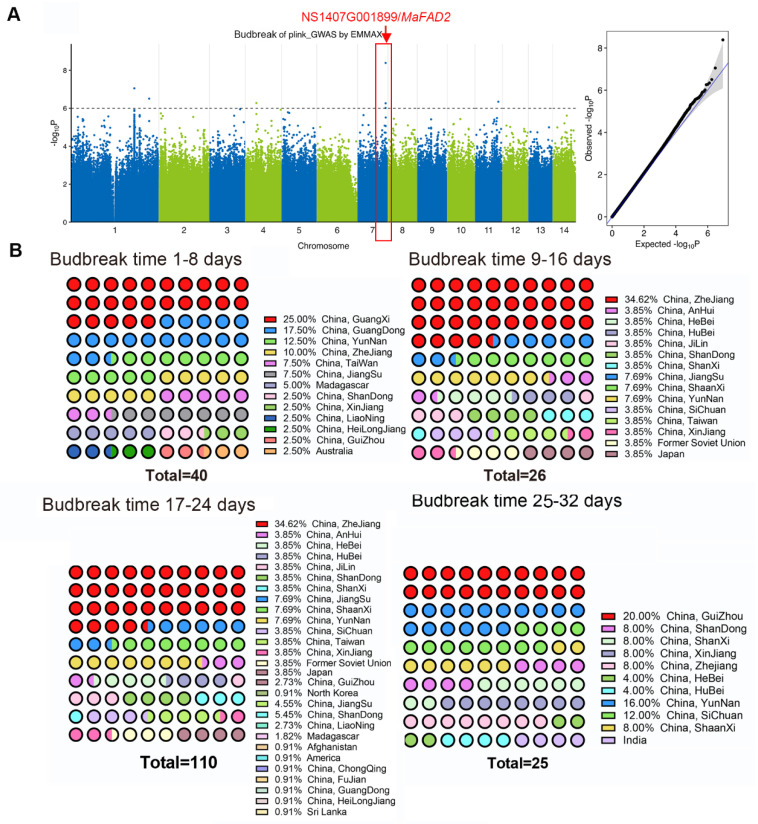
GWAS for budbreak date in 2023. (**A**) Genome–wide Manhattan plot of the Compressed Mixed Linear Model. The significance threshold is denoted by the black dashed line. The 14 mulberry chromosomes are plotted against the negative base-10 logarithm of the association *p*-value. The right side is a quantile–quantile plot (Q-Q plot), which represents the distribution of the actual observed *p*-values and the expected *p*-values under the null hypothesis of no association. The X and Y axes represent the −log_10_ *p*-values of each SNP. The predicted line is a dashed line at a 45° angle from the origin. (**B**) Frequency statistics on the distribution of the budbreak date in 201 natural populations of mulberry trees.

**Figure 2 ijms-25-13341-f002:**
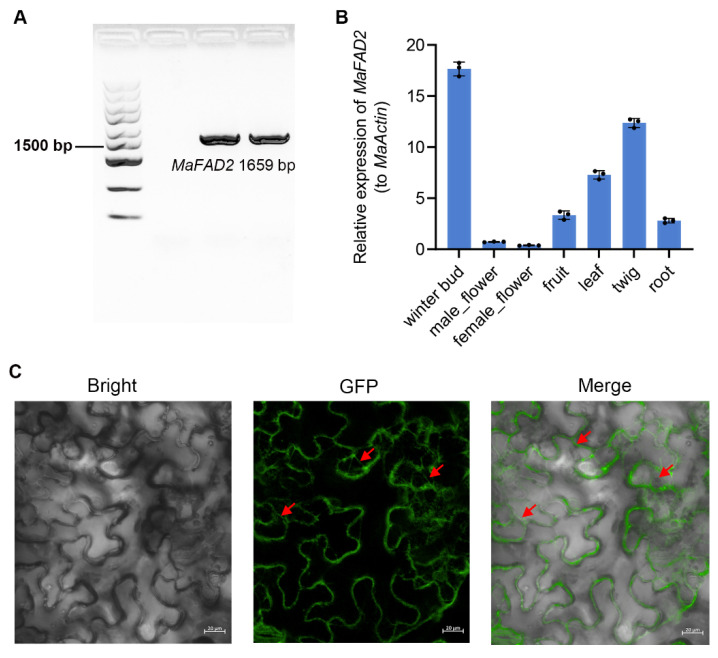
Functional study of the *MaFAD2* gene. (**A**) Cloning of the coding region of *MaFAD2*. The horizontal line indicates the size of the 1500 bp marker. (**B**) Detection of *MaFAD2* expression in different tissues of mulberry. (**C**) Subcellular localization of MaFAD2. GFP fluorescence signals indicate MaFAD2 expression. The red arrows illustrate endoplasmic reticulum localization signals.

**Figure 3 ijms-25-13341-f003:**
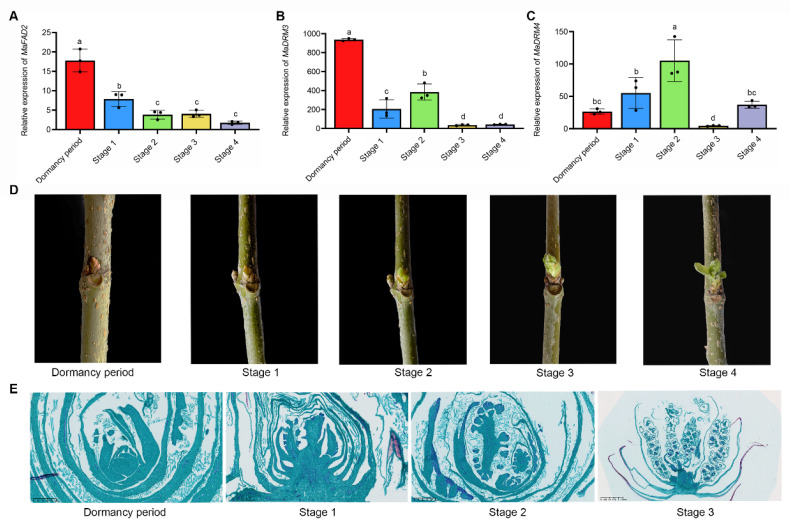
*MaFAD2* gene expression decreased with mulberry budbreak. (**A**–**C**) Changes in gene expression with different stages of mulberry bud break; (**B**,**C**) are positive controls. Different letters indicate a significant difference (*p* < 0.05). Data are represented as mean ± SD. (**D**) Schematic diagram of the different stages of mulberry bud break. (**E**) Safranin O-Fast Green-stained sections from different stages of mulberry bud break.

**Figure 4 ijms-25-13341-f004:**
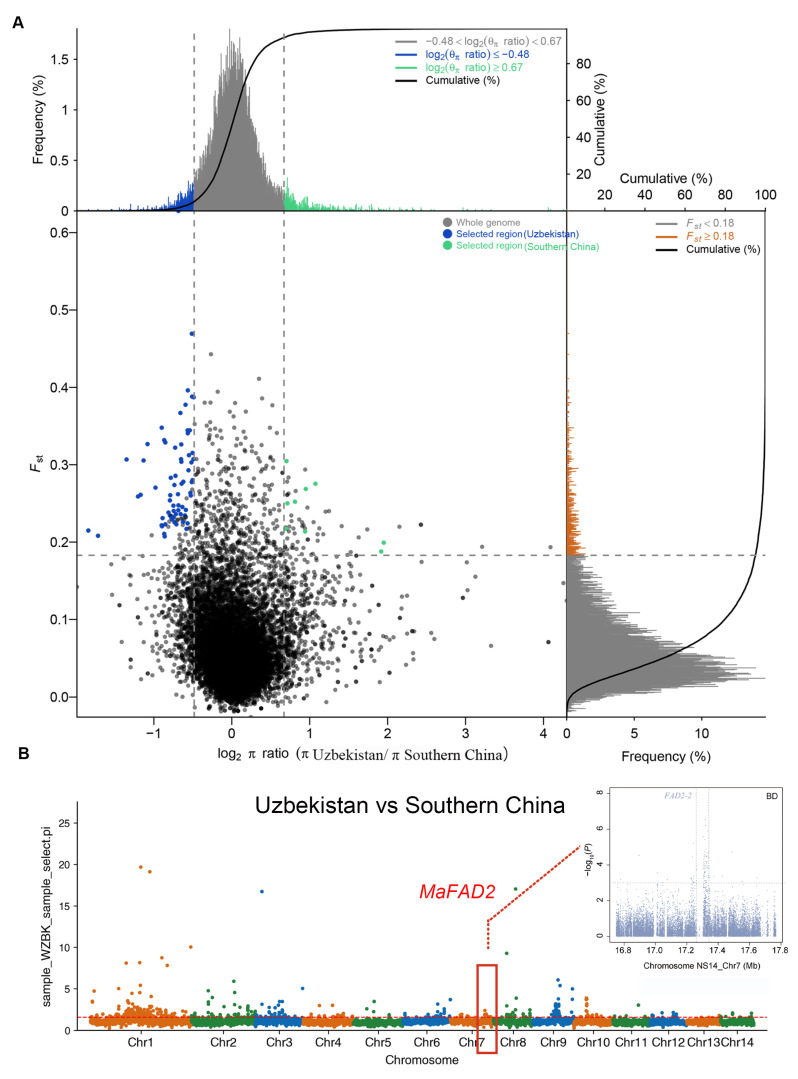
Genome-wide analysis of selective sweeps during *Morus alba* budbreak date. (**A**) Distribution of population differentiation (Fst) and π ratios (log2(π Uzbekistan/π Southern China)) between Uzbekistan and southern China using a 50 kb sliding window and 25 kb steps. The horizontal axis represents the ratio of θ_π_ (Uzbekistan/southern China), and the vertical axis represents the Fst values, corresponding to the frequency distribution diagram on the top and the right side, respectively. The scatter plot in the middle represents the corresponding Fst and θ_π_ ratios within different windows. The top blue and green areas represent the top 5% regions selected by θ_π_, while the red area represents the top 5% region selected by Fst. The middle blue and green areas are the intersection of Fst and θ_π_, which are the candidate sites. (**B**) Genome-wide selective signals between Uzbekistan and southern China *Morus alba*. The horizontal dashed lines indicate the cutoffs.

**Figure 5 ijms-25-13341-f005:**
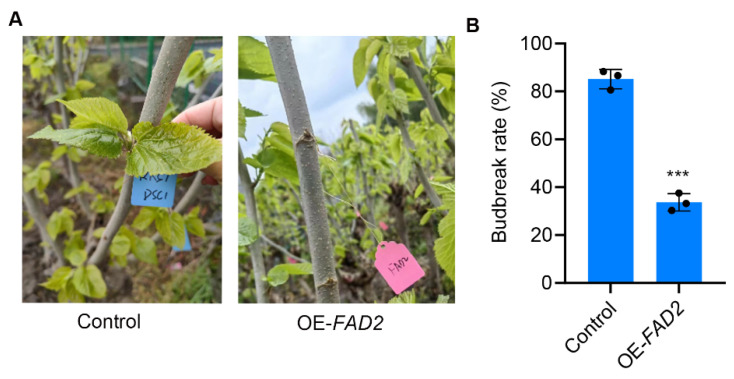
Overexpressing the *MaFAD2* gene in the winter buds of mulberry trees can delay their budbreak. Budbreak of mulberry trees after overexpression of the *MaFAD2* gene (**A**) and budbreak statistics (**B**). Data are presented as mean ± SD; asterisks indicate a significant difference (*** *p* < 0.001, Student’s *t*-test).

## Data Availability

The data presented in this study are available upon request from the corresponding author. They are not publicly available because some results have not yet been organized and published.

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
