# Peer review of "The Role of MaFAD2 Gene in Bud Dormancy and Cold Resistance in Mulberry Trees (Morus alba L.)"

_ijms, 2024, doi:10.3390/ijms252413341_

Round 1
Reviewer 1 Report
Comments and Suggestions for Authors
Interesting manuscript abouit the moleuclar determinism of bud formancy in Morus alba. However a major revision is required before publication:
Objectives (lines 92-107) of the work must be sumarized only indicating the key new aportations of the work.
Legend of Figure 2 in page 5 must be completed. The figuers must be self-redeables with the incorporated legend.
Discussion section must be completed wint mor refernces about bud dormancy in mulberry or related species.
A new section of cu¡onclusions (at this moment are very weak, lines 269-274) must be imporved indicating main implications of the obtained results from a production and breeding point of view.
Description od qPCR methodology (lines 308-320) must be improved adding the refernce genes assayed and the satatistical analysis performed.
Agrobacterium protocol (lines 363-370) must be completed indicating a detailed description. Results of this assay in section 2.5 must be also completed indicating the ersults of the assayed regeneration and acclimatation protocol.
Author Response
Comments and Suggestions for Authors
Interesting manuscript abouit the moleuclar determinism of bud formancy in Morus alba. However a major revision is required before publication:
- Objectives (lines 92-107) of the work must be sumarized only indicating the key new aportations of the work.
Response1: Thank you for your advice. we have streamlined this section to highlight the most important findings and key new contributions of the research. (Lines 85-95)
- Legend of Figure 2 in page 5 must be completed. The figuers must be self-redeables with the incorporated legend.
Response2: Thank you for your suggestions. we have expanded the legends for Figure 2 to make it as detailed as possible for the convenience of the readers' understanding. The specific expressions are as follows: “Figure 2. Functional study of the MaFAD2 gene. (A) Cloning of the coding region of MaFAD2. The arrow indicates the size of the 1500bp marker. (B) Detection of MaFAD2 expression in different tissues of mulberry. (C) Subcellular localization of MaFAD2. GFP fluorescence signals indicate MaFAD2 expression. The red arrows illustrate endoplasmic reticulum localization signals. ” (Lines 517-520, 558-561)
- Discussion section must be completed wint mor refernces about bud dormancy in mulberry or related species.
Response3: Thank you for the reminder, we have already supplemented the relevant literature on tree dormancy in some parts of the discussion. (Lines 191 and 192)
- A new section of cu¡onclusions (at this moment are very weak, lines 269-274) must be imporved indicating main implications of the obtained results from a production and breeding point of view.
Response4: Following your suggestions, we have made revisions to the discussion section. We have conducted a detailed discussion and analysis in terms of breeding and practical production, hoping to meet your requirements. (Lines 222-255)
- Description od qPCR methodology (lines 308-320) must be improved adding the refernce genes assayed and the satatistical analysis performed.
Response5: According to your suggestion we have supplemented the internal reference genes and analytical methods in the Methods section and introduced the corresponding references as follows: “The MaActin gene was used as an internal control for qPCR, and relative expression levels were calculated using the comparative Cq (quantification cycle) method.” (Lines 301-303)
- Agrobacterium protocol (lines 363-370) must be completed indicating a detailed description. Results of this assay in section 2.5 must be also completed indicating the ersults of the assayed regeneration and acclimatation protocol.
Response6: We apologize for the lack of detail in the methodology section, and we have supplemented the section “4.7 Agrobacterium-mediated transformation into winter bud of Mulberry tree” with additional information on the method of injecting winter buds. The supplemental content is as follows: “In this study, the experimental strain used was the Agrobacterium tumefaciens strain (GV-pBI121) carrying the β-glucuronidase gene (GUS) and the kanamycin resistance (Kan) reporter gene, provided by Dr. Guoxin Shen from the Zhejiang Academy of Agricultural Sciences. The GUS gene in this strain is controlled by the 35S promoter of the cauliflower mosaic virus. Initially, the preserved bacterial strain was inoculated onto LB medium containing kanamycin (100 mg/L) and rifampicin (50 mg/L), and cultured in a 28℃ constant temperature incubator. After the formation of single colonies, they were transferred to liquid LB medium containing kanamycin (50 mg/L) and rifampicin (50 mg/L), and further cultured at 28℃ with a shaking speed of 200 r/min until the bacterial suspension reached an OD600 of 0.5. The cultured Agrobacterium solution was centrifuged at room temperature for 15 min (4000 r/min), the supernatant was discarded, and the Agrobacterium was resuspended in an equal volume of infection buffer (MES 0.25g/L, Myo-inositol 0.10g/L, MS with Gamborg Vitamins 2.22g/L, L-glutamine 0.20g/L, D-galactose 1.80g/L, pH 5.0). ” and “Using a 1 mL syringe, the Agrobacterium infection solution was injected into the winter buds of mulberry trees, with 0.1 mL per bud. After the injection, the branches were wrapped with aluminum foil and continued to be cultured.” (Lines 354-372)
Additionally, regarding the section "2.5 Overexpression of MaFAD2 significantly postponed the timing of bud break," we have supplemented the experimental details as follows: “We conducted three independent biological replicates, with 10 mulberry trees injected in each replicate. After 10 days, the germination was statistically assessed.” and “We can clearly observe that the budbreak rate of the winter buds from mulberry trees overexpressing MaFAD2 is half that of the control group. Additionally, the vast majority of the winter buds in the control group developed normally, while a small portion did not budbreak properly, and we cannot rule out the influence of environmental or injection factors”. (Lines 173-175, 176-181)

Reviewer 2 Report
Comments and Suggestions for Authors
The Role of MaFAD2 Gene in Bud Dormancy and Cold Re-1 sistance in Mulberry Trees (Morus alba L.)
Mengjie Zhao, Gaoxing Zhou, Peigang Liu, Zhifeng Wang, Lu Yang, Tianyan Li, Valiev Sayfiddin Tojid- dinovich , Nasirillayev Bakhtiyar Ubaydullayevich, Ismatullaeva Diloram Adilovna, Khudjamatov Safarali Khasanboy ugl , Yan Liu, Zhiqiang Lv, Jia Wei, and Tianbao Lin.
The draft describes the study of the association between bud break timing (related with the loss of the dormancy) and cold resistance (in varieties with different resistance to cold), was studied using a genome-wide association study approach in Morus alba L. In which, the candidate FAD2 (gene that encodes a Delta 12 fatty acid desaturase) presented the strongest association, according to, the analysis of Genetic association tests (Manhattan plot) and enrichment of association signal (quantile–quantile plot). The authors verify the association of this gene with the loss of dormancy by analyzing the expression profiles in tissues with different stages of bud break. Additionally, the authors associating the expression profile of the FAD2 genes with that of widely reported genes related to it bud dormancy. Complementing the analysis with gene overexpression with the time of bud break.
Overall, the manuscript well structured, and the flow is good, the methods are adequate. It is evident that the manuscript would benefit from additional. Therefore, I suggest the following to the authors:
1.-The authors do not provide information in the materials and methods section, of the draft of the high-SNP-density dataset used in this study. Or the whole-genome resequencing data of the 201 natural ecotypes of mulberry available.
2.-lines 305-306.
Rather of : "Potential candidate SNP were identified through the significance of association (P-value) [33]."
Could be: "A significance threshold of p ≤ 10−6 (i.e., −log10 p ≥ 6) was used to identify significant associations [33]."
3.- The authors mention in the results section (lines 139-140) "Initially, we cloned the coding sequence of MaFAD2 and found it to be approximately 1700 base pairs in length (Figure 2A). "While, in la section "4.4. Subcellular localization analysis of proteins", briefly mention the cloning and sequencing protocol.
In it draft, they do not disclose the nucleotide sequence or a data accession number. This information would benefit the work.
for example: Two genes were identified from mesocarp tissue of olive fruit from the Picual and Arbequina, in tissue incubated at 15 °C, transcript levels was observed with a temporary induction of slight for FAD2-1 and intense for FAD2-2.
With the nucleotide sequence available, it is possible to accurately reference the identified gene for future analysis.
Hernández, M. L., Padilla, M. N., Sicardo, M. D., Mancha, M., & Martínez-Rivas, J. M. (2011). Effect of different environmental stresses on the expression of oleate desaturase genes and fatty acid composition in olive fruit. Phytochemistry, 72(2-3), 178-187.
4.-Finally, in section "4.7. Agrobacterium-mediated transformation into winter bud of Mulberry tree".
The authors mention (lines 364-365) "The methods of injection and transformation refer to the article by Lin et al. 364 (2018)[31]. "
In google scholar, I was unable to find reference 31. Since access to the article is complicated, it is recommended that the authors describe in more detail the transformation protocol and the transformed tissue testing protocol.
Author Response
Comments and Suggestions for Authors
The Role of MaFAD2 Gene in Bud Dormancy and Cold Re-1 sistance in Mulberry Trees (Morus alba L.)
Mengjie Zhao, Gaoxing Zhou, Peigang Liu, Zhifeng Wang, Lu Yang, Tianyan Li, Valiev Sayfiddin Tojid- dinovich , Nasirillayev Bakhtiyar Ubaydullayevich, Ismatullaeva Diloram Adilovna, Khudjamatov Safarali Khasanboy ugl , Yan Liu, Zhiqiang Lv, Jia Wei, and Tianbao Lin.
The draft describes the study of the association between bud break timing (related with the loss of the dormancy) and cold resistance (in varieties with different resistance to cold), was studied using a genome-wide association study approach in Morus alba L. In which, the candidate FAD2 (gene that encodes a Delta 12 fatty acid desaturase) presented the strongest association, according to, the analysis of Genetic association tests (Manhattan plot) and enrichment of association signal (quantile–quantile plot). The authors verify the association of this gene with the loss of dormancy by analyzing the expression profiles in tissues with different stages of bud break. Additionally, the authors associating the expression profile of the FAD2 genes with that of widely reported genes related to it bud dormancy. Complementing the analysis with gene overexpression with the time of bud break.
Overall, the manuscript well structured, and the flow is good, the methods are adequate. It is evident that the manuscript would benefit from additional. Therefore, I suggest the following to the authors:
1.-The authors do not provide information in the materials and methods section, of the draft of the high-SNP-density dataset used in this study. Or the whole-genome resequencing data of the 201 natural ecotypes of mulberry available.
Response1: Thank you for your valuable feedback. We appreciate your concerns regarding the lack of information on the high-SNP-density dataset and the whole-genome resequencing data of the 201 natural ecotypes of mulberry. Due to ongoing and unpublished research, we are unable to publicly share the complete dataset at this time. However, we have extracted the relevant data to support our findings and included a table with the FAD2-related data as Supplementary File 1 in the revised manuscript. We have also made revisions to clarify this situation in the manuscript. We hope this addresses your concerns.
2.-lines 305-306. Rather of : "Potential candidate SNP were identified through the significance of association (P-value) [33]." Could be: "A significance threshold of p ≤ 10−6 (i.e., −log10 p ≥ 6) was used to identify significant associations [33]."
Response2: This is a very good suggestion, and we have changed the expression in the manuscript to “A significance threshold of P-value ≤ 10−6 (i.e., −log10P≥ 6) was used to identify significant associations” at your suggestion. (Lines 286-288)
3.- The authors mention in the results section (lines 139-140) "Initially, we cloned the coding sequence of MaFAD2 and found it to be approximately 1700 base pairs in length (Figure 2A). "While, in la section "4.4. Subcellular localization analysis of proteins", briefly mention the cloning and sequencing protocol.
In it draft, they do not disclose the nucleotide sequence or a data accession number. This information would benefit the work.
for example: Two genes were identified from mesocarp tissue of olive fruit from the Picual and Arbequina, in tissue incubated at 15 °C, transcript levels was observed with a temporary induction of slight for FAD2-1 and intense for FAD2-2.
With the nucleotide sequence available, it is possible to accurately reference the identified gene for future analysis.
Hernández, M. L., Padilla, M. N., Sicardo, M. D., Mancha, M., & Martínez-Rivas, J. M. (2011). Effect of different environmental stresses on the expression of oleate desaturase genes and fatty acid composition in olive fruit. Phytochemistry, 72(2-3), 178-187.
Response3: We couldn't agree with you more. We have now added the MaFAD2 gene ID as well as coding region sequence information to the manuscript (Supplementary file 1). And we have modified and added some statements at the corresponding positions in the manuscript. (Lines 188 and 121)
4.-Finally, in section "4.7. Agrobacterium-mediated transformation into winter bud of Mulberry tree".
The authors mention (lines 364-365) "The methods of injection and transformation refer to the article by Lin et al. 364 (2018)[31]. "
In google scholar, I was unable to find reference 31. Since access to the article is complicated, it is recommended that the authors describe in more detail the transformation protocol and the transformed tissue testing protocol.
Response4: We apologize for our oversight in not providing the necessary information earlier. This is a Chinese-language journal. We have now corrected the references and included the DOI number for this article, which should help you locate the literature.
Additionally, we have supplemented the specific method of Agrobacterium injection at the corresponding location in the methods section, that is “In this study, the experimental strain used was the Agrobacterium tumefaciens strain (GV-pBI121) carrying the β-glucuronidase gene (GUS) and the kanamycin resistance (Kan) reporter gene, provided by Dr. Guoxin Shen from the Zhejiang Academy of Agricultural Sciences. The GUS gene in this strain is controlled by the 35S promoter of the cauliflower mosaic virus. Initially, the preserved bacterial strain was inoculated onto LB medium containing kanamycin (100 mg/L) and rifampicin (50 mg/L), and cultured in a 28℃ constant temperature incubator. After the formation of single colonies, they were transferred to liquid LB medium containing kanamycin (50 mg/L) and rifampicin (50 mg/L), and further cultured at 28℃ with a shaking speed of 200 r/min until the bacterial suspension reached an OD600 of 0.5. The cultured Agrobacterium solution was centrifuged at room temperature for 15 min (4000 r/min), the supernatant was discarded, and the Agrobacterium was resuspended in an equal volume of infection buffer (MES 0.25g/L, Myo-inositol 0.10g/L, MS with Gamborg Vitamins 2.22g/L, L-glutamine 0.20g/L, D-galactose 1.80g/L, pH 5.0). ” and “Using a 1 mL syringe, the Agrobacterium infection solution was injected into the winter buds of mulberry trees, with 0.1 mL per bud. After the injection, the branches were wrapped with aluminum foil and continued to be cultured.” (Lines 354-372 and 495)

Round 2
Reviewer 1 Report
Comments and Suggestions for Authors
Authors have revised correctly the manuscript. Only minor changes are required before pubblication:
Please increase quality of Figures 1 and 4 increasing font size.
Author Response
Comment1:Please increase quality of Figures 1 and 4 increasing font size.
Response1:Thank you for your advice. We have adjusted the font size of Figures 1 and 4 for readability. And it has been updated in the manuscript.